# Real-time $^{31}$P NMR reveals different gradient strengths in polyphosphoester copolymers as potential MRI-traceable nanomaterials

Timo Rheinberger [1], Ulrich Flögel [2], Olga Koshkina[1] & Frederik R. Wurm[1✉]

Polyphosphoesters (PPEs) are used in tissue engineering and drug delivery, as polyelectrolytes, and flame-retardants. Mostly polyphosphates have been investigated but copolymers involving different PPE subclasses have been rarely explored and the reactivity ratios of different cyclic phospholanes have not been reported. We synthesized binary and ternary PPE copolymers using cyclic comonomers, including side-chain phosphonates, phosphates, thiophosphate, and in-chain phosphonates, through organocatalyzed ring-opening copolymerization. Reactivity ratios were determined for all cases, including ternary PPE copolymers, using different nonterminal models. By combining different comonomers and organocatalysts, we created gradient copolymers with adjustable amphiphilicity and microstructure. Reactivity ratios ranging from 0.02 to 44 were observed for different comonomer sets. Statistical ring-opening copolymerization enabled the synthesis of amphiphilic gradient copolymers in a one-pot procedure, exhibiting tunable interfacial and magnetic resonance imaging (MRI) properties. These copolymers self-assembled in aqueous solutions, 31P MRI imaging confirmed their potential as MRI-traceable nanostructures. This systematic study expands the possibilities of PPE-copolymers for drug delivery and theranostics.

[1] Sustainable Polymer Chemistry (SPC), Department of Molecules and Materials, MESA+ Institute for Nanotechnology, Faculty of Science and Technology, University of Twente, P.O. Box 217, 7500 AE Enschede, Netherlands. [2] Department of Molecular Cardiology, Experimental Cardiovascular Imaging, Heinrich-Heine-University, Düsseldorf, Germany. ✉email: f.r.wurm@utwente.nl

Polyphosphoesters (PPEs) represent a platform of versatile polymers that have been investigated for various applications, ranging from biomedicine (e.g., drug delivery and tissue engineering) to materials science (flame retardants, electrolytes)[1,2]. PPEs are accessible as hydrophobic but, more importantly, also hydrophilic polymers via classical polycondensations, olefin metathesis, and ring-opening polymerization (ROP). ROP leads to well-defined polymers and is typically catalyzed by transesterification catalysts relying on metals, e.g., stannous octanoate, or more recently, various organocatalysts, such as 1,8-Diazabicyclo[5.4.0]undec-7-ene (DBU)[3,4]. As the use of metals during the process can lead to unwanted metallic residues in the product resulting in unwanted conductivity and health concerns, the use of organocatalysts also increased in the synthesis of PPEs, similar to the ROP of conventional cyclic esters, like lactide. In previous reports on the organocatalyzed ROP of cyclic phosphoesters, different organocatalysts have been reported, or the use of additional compounds, such as urea-derivates, that control the reactivity of the chain ends.

For the PPE platform, a series of subclasses exist (Fig. 1a), which exhibit different properties, such as solubility, degradation, and thermal and mechanical properties. Copolymers of these subclasses would offer the possibility of combining these properties. However, the copolymerization of different PPE subclasses has been unexplored so far. Here, we systematically study the copolymerization of monomers belonging to the different PPE subclasses (side- and in-chain phosphonates, phosphates, and thiophosphates) using organocatalysis.

Copolymerization, generally, can provide access to a range of copolymer topologies, such as block-, random, gradient, and tapered copolymers, allowing to tune various properties, such as hydrophilicity and functionality[5,6]. In ROP, the choice of organocatalyst and reaction conditions control the reaction kinetics for each of the monomer subclasses, i.e., phospholanes, shown in Scheme 1[4,7]. So far, phosphates are the most explored PPE-

subclass, with ca. 30 different 1,3,2-dioxaphospholane monomers (i.e., cyclic phosphates) reported; their statistical copolymerization typically led to random copolymers[4,8]. The development of side-chain polyphosphonates allowed for further tailoring of the polymer properties, e.g., amphiphilic gradient copolymers were reported through copolymerization of different cyclic phosphonates[9]. Both phosphates and phosphonates can also be copolymerized with common monomers like lactides[10,11], and other lactones[12] or carbonates[13] using metal- or organocatalysis.

Here, we developed general protocols for the organocatalyzed copolymerization of the four different PPE subclasses shown in Fig. 1. We followed the copolymerizations via real-time $^1$H/$^{31}$P NMR spectroscopy and were able to elucidate the formation of gradient copolymers with different gradient profiles, depending on the comonomer combination and the type of organocatalyst. Reactivity ratios for all comonomer sets have been calculated by different nonterminal models and are reported for binary and ternary copolymers. Copolymerization of phosphates and phosphonates resulted in gradient copolymers with different gradient strengths. Different behavior was observed for in-chain polyphosphonates and polythiophosphates, which both are known to homopolymerize slower than the other subclasses[14,15]; here, statistical copolymerization resulted in block-like structures in one step. Changing the properties is important for future applications: for example, polythiophosphates are more hydrophobic and redox responsive[15]. For these comonomer sets, different gradient profiles have been identified, and their interfacial properties have been investigated using the spinning drop technique. Further, such gradient copolymers assembled into nanostructures in water, which were analyzed by light scattering, and thus they gave a possibility to trace such biomedical polymers with magnetic resonance imaging (MRI). MRI of dispersions is reported as well, indicating a strong influence on the gradient structure and assembly on the signal-to-noise ratio in MRI. Together, these PPE copolymers can be tailored in terms of hydrophobicity,

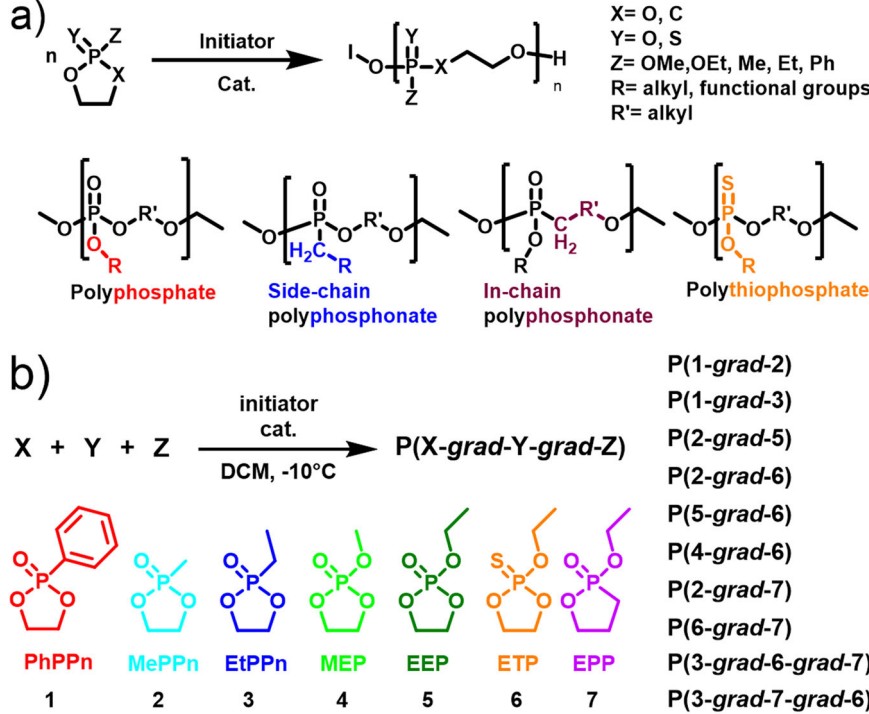

**Fig. 1 Synthesis of polyphosphoester copolymers using organocatalyzed ring-opening polymerization of cyclic polyphosphoester monomers. a** General structures of the polyphosphoester subclasses (phosphates, side-chain phosphonates, in-chain phosphonate, thiophosphate), **b** synthesis of various gradient copolymers from different polyphosphoester classes (side-chain phosphonates (1–3), phosphates (4, 5), thiophosphate (6), in-chain phosphonate (7).

**Table 1 Overview of PPE copolymers, synthesis parameters, and molar mass data (all polymerizations were performed in DCM at −10 °C in an NMR tube directly in the spectrometer).**

| # | Copolymers | Reaction time/h | Catalysts | Total conv.[a] | DP$_n$ per monomer | Monomer ratio[a] | Đ[b] | $M_n$ NMR/g mol[c] | $T_g$[d] |
|---|---|---|---|---|---|---|---|---|---|
| P1 | P(PhPPn-*grad*-MePPn) | 22 | DBU | 93% | 48/41 | 55/45 | 1.35 | 13.9k | −27.9 |
| P2 | P(PhPPn-*grad*-EtPPn) | 24 | DBU | 83% | 48/43 | 55/45 | 1.1 | 14.8k | −32.7 |
| P3 | P(EtPPn-*grad*-ETP) | 26 | DBU/TU | 83% | 47/19 | 70/30 | 1.19 | 9.7k | −48.0 |
| P4 | P(EtPPn-*grad*-EEP) | 19 | DBU/TU | 97% | 90/71 | 52/48 | 1.1 | 23.1k | −53.1 |
| P5 | P(EEP-*grad*-ETP) | 30 | DBU/TU | 86% | 47/33 | 59/41 | 1.23 | 12.2k | n.D. |
| P6 | P(MEP-*grad*-ETP) | 24 | DBU/TU | 83% | 53/24 | 70/30 | 1.23 | 11.4k | −52.7 |
| P7 | P(EtPPn-*grad*-EPP) | 68 | DBU/TU | 75% | 53/29 | 64/36 | 1.12 | 11.7k | −52.7 |
| P8 | P(ETP-*grad*-EPP) | 122 | DBU/TU | 72% | 39/28 | 60/40 | 1.99 | 11.1k | −47.1 |
| P9 | P(EtPPn-*grad*-ETP-*grad*-EPP) | 125 | DBU/TU | 73% | 51/29/13 | 54/33/14 | 1.12 | 13.9k | −48.5 |
| P10 | P(EtPPn-*grad*-EPP-*grad*-ETP) | 45 | DBU/ TrisUrea | 98% | 50/32/25 | 47/30/23 | 1.47 | 16.0k | −51.1 |

[a]Determined from $^{31}$P NMR measurements.
[b]Determined via GPC in DMF (0.1 M LiCl, at 50 °C) vs. polystyrene standards (elugrams summarized in Fig. S1).
[c]Determined from $^1$H NMR measurements in CDCl$_3$ after workup.
[d]Determined from Differential Scanning Calorimetry (DSC) using the second heating ramp.

degradation[16], and further functionalities, making them interesting candidates for new degradable theranostics[17].

## Results and discussion

To explore the copolymerization behavior of the different PPE subclasses, we conducted statistical organocatalysed ring-opening copolymerizations of two or three different monomer types (Fig. 1b). For the ROP of phospholanes, previously different organocatalysts had been used, such as 1,8-Diazabicyclo[5.4.0] undec-7-ene (DBU) and 1,5,7-Triazabicyclo[4.4.0]dec-5-en (TBD), alone or in combination with urea-derivatives, depending on the reactivity of the respective phospholane[4,8]. Here, we focused on DBU for the copolymerization of the side-chain phosphonates, as no transesterification had been reported under these conditions[18]. For the other monomer classes, we used the combination of N-Cyclohexyl-N′-(3,5-bis(trifluoromethyl)phenyl) thiourea (TU) and DBU, to prevent transesterification and enhance the reaction rates[19]. We further studied the 1,1′,1″-(nitrilotris(ethane-2,1-diyl))tris(3-(3,5-bis(trifluoromethyl)phenyl)urea) (Tris Urea) in combination with DBU, to increase the reactivity of the in-chain phosphonates which are known to have a very low reactivity[14]. All copolymerizations were performed in DCM at −10 °C directly in an NMR tube in the spectrometer to gather all kinetics data for each comonomer set and calculate the reactivity ratios and the resulting microstructure (comonomer distribution) of the resulting copolymers (see below). In all cases, we obtained well-defined copolymers with monomodal narrow to moderate molar mass dispersity, indicating the controlled nature of the copolymerizations (SI Fig. S1, Table 1). We targeted degrees of polymerization of ca. 100, resulting in molar masses of up to 23 kg/mol, as shown by NMR spectroscopy and gel permeation chromatography (GPC), but this should not be regarded as a limitation.

We selected the reaction temperature of −10 °C to reduce side reactions, for example, transesterification, based on previous reports (it also has to be noted that the copolymerization was conducted directly in the NMR spectrometer without any stirring, which could lead to some broadening of the molar mass dispersity)[8]. Further, the copolymerization kinetics are slowed down to a time scale, allowing to study the reaction kinetics, i.e., comonomer consumption and the formation of the copolymers, to be followed by real-time $^{31}$P NMR spectroscopy.

**Binary copolymers.** Figure 2 exemplarily shows the $^{31}$P NMR data obtained for the copolymerization of PhPPn and MePPn

followed for 22 h ($^1$H NMR, see Fig. S3). The resonance of monomer 1 (at 36.7 ppm, red signal in Fig. 2) and 2 (at 48.7 ppm, gray signal in Fig. 2) decrease during the reaction, while the polymer resonances at 19.9 ppm (green signal in Fig. 2) and 30.4 ppm (blue signal in Fig. 2), respectively appear, confirming the incorporation of both monomers into a copolymer structure.

To calculate the microstructure of the copolymers, the integral values of each comonomer during the respective copolymerization were plotted overtime against the total monomer conversion (Fig. 3a). From these raw data obtained from the *real-time* $^{31}$P NMR measurements, we calculated the reactivity ratios ($r$) for the respective comonomer set. We used nonterminal models, Jaacks[20], BSL (Beckingham, Sanoja, and Lynd)[21], and Frey (ideal integrated model)[22]. To prevent potential errors in the fitting procedure, we combined the results from three of these models, ensuring reliable reactivity ratios from all data fitting (Figs. 3 and Suppl. Fig.). In all cases, the three models and fittings resulted in relatively similar values, which we present as single and averaged values for the reactivity ratios of each comonomer pair. The reactivity ratios from the ideal integrated model were used to visualize the comonomer distribution by plotting the polymer fraction vs. monomer conversion (Fig. 3e). Based on these results, we performed Monte-Carlo simulations to demonstrate the microstructure of the copolymer chain (Fig. 3f; for the visualization, 10 individual polymer chains are plotted with a theoretical dispersity set to unity to exclude chain length effects).

For the copolymerization of 1 and 2, $r(\text{PhPPn}) = 4.9 \pm 0.3$ and $r(\text{MePPn}) = 0.21 \pm 0.01$ were obtained by this approach; the experimental data of up to 60% total monomer conversion were used for the calculations as recommended previously for other copolymerizations; Fig. 3b–d)[21].

Gleede et al. defined the gradient strength of copolymers based on the difference of the reactivity ratios $r_1$ and $r_2$ of two comonomers[23]. Another more simplified definition of the gradient strength in copolymers uses the difference $\Delta r$ between $r_1$ and $r_2$ (assuming ideal polymerization with $r_1 \cdot r_2 = 1$), leading to the following definitions:

Soft gradient for $0 < \Delta r \le 1.5$
Medium gradient for $1.5 < \Delta r \le 7.5$
Hard gradient for $7.5 < \Delta r \le 25$
Block(-like) for $25 < \Delta r$

This definition of gradient strength was used to describe our prepared copolymers which all showed a certain gradient structure; thus, the copolymer structure obtained from 1 and 2 was classified as a medium gradient[23].

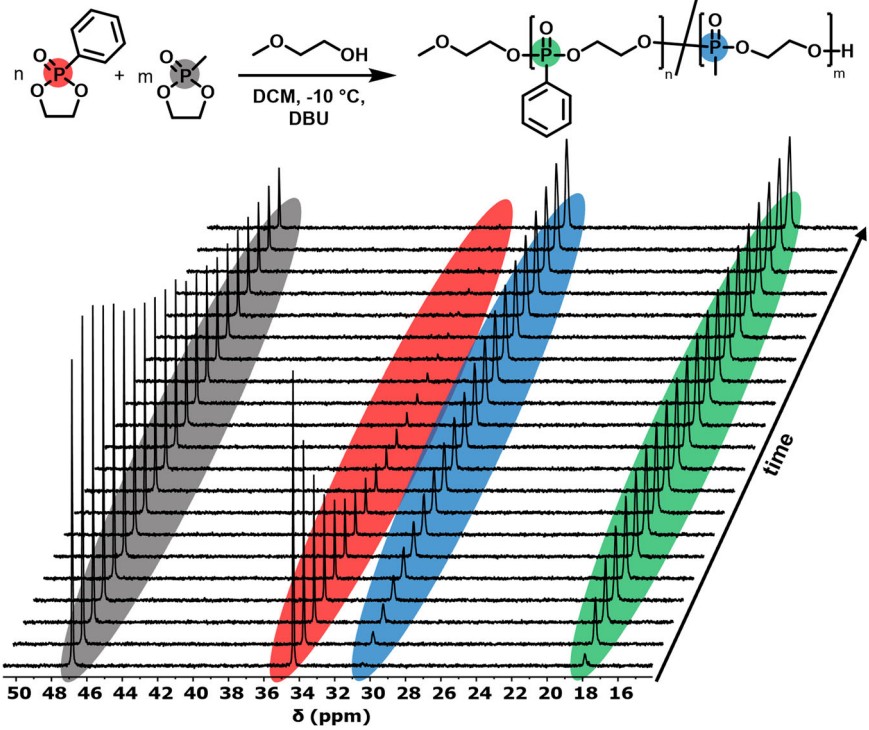

**Fig. 2 Real-time ³¹P{H} NMR spectroscopy gives access to reactivity ratios during copolymerization.** PhPPn **1** and MePPn **2** (entry **P1**, Table 1) were copolymerized in DCM (4 M) at −10 °C. Shown is an overlay of ³¹P NMR spectra with highlighted resonances of the monomers before and after incorporation into the copolymer (the interval between the first 10 spectra was set to 5.7 min, for the spectra 10–20 to 11.3 min).

For the other copolymerizations of the different monomer combinations with the respective organocatalysts (see Table 1, P2–P8, Fig. 4), the analyses results indicate different comonomer reactivities under the applied conditions, which resulted in copolymers with different gradient strengths (Fig. 4, Table 1, Figs. S4–S10). The reactivity ratios of all monomers all showed good fits (Figs. S4–S10 with $R^2 > 0.9$) and small standard deviation (Fig. 4). The corresponding copolymer structures are presented as monomer fraction composition plotted against the total conversion and via the Monte Carlo simulation of ten individual copolymer chains.

Changing the electron density of phosphorus within the class of side-chain polyphosphonates allows to tailor of the gradient strengths. When hydrophilic MePPn (2) in pair with rather hydrophobic PhPPn (1) is copolymerized, a copolymer with a medium gradient ($\Delta r = 5$) was obtained (P1). Differently, the copolymerization of PhPPn with the ethyl derivative (EtPPn, 3) resulted in a hard gradient ($\Delta r = 24$, P2). These results indicate that a lower electron density at P, which results from the inductive effect of the side-chain, leads to the faster reaction kinetics of methyl-PPn compared to ethyl-PPn[18].

When polythiophosphates are used as hydrophobic block[15], the copolymerization of ETP with EtPPn resulted in copolymers with a hard gradient ($\Delta r = 16$) P(EtPPn-grad-ETP) (P3) with amphiphilic properties. Here, sulfur has a diffuse electron density, favoring higher electron density at phosphorus. Consequently, a hard gradient was obtained.

The copolymerization of the phosphate EEP and the phosphonate EtPPn produced a medium gradient structure ($\Delta r = 3$) P(EtPPn-grad-EEP) (P4) with a double hydrophilic structure. It appears that the electron density alone is not able to explain this result, suggesting that the combination of the electron environment of phosphorus, with ring tension and potentially other factors, govern the kinetics. Based on this reaction, similar

reactivity ratios of aliphatic phosphate and phosphonate monomers can be predicted (since also monomers with longer n-alkyl side chains have been reported to produce random copolymers)[24]. Based on the kinetics measured in the above experiments, a combination of EEP and ETP should yield a hard gradient. Indeed, the copolymerization yields P(EEP-grad-ETP) (P5) with a hard gradient and $\Delta r = 10$ (close to a medium gradient), proving this hypothesis. MEP is faster than EEP, and therefore the copolymerization with ETP results in P(MEP-grad-ETP) (P6) with a greater $\Delta r = 18$. Again, the replacement of oxygen by sulfur slows down the polymerization, leading to an increase in the gradient strength.

Comparing the side- and in-chain polyphosphonate, the ring strain plays a key role in kinetics and the resulting topology. In-chain PPn polymerizes slower than side-chain PPn[14]. Hence, copolymerizing EPP with EtPPn results in a double hydrophilic block-like structure ($\Delta r = 26$) P(EtPPn-grad-EPP) (P7). By using the hydrophobic ETP as a comonomer, we achieved P(ETP-grad-EPP) (P8) with a medium gradient ($\Delta r = 5$) with amphiphilic properties, as expected from the above kinetic measurement.

The resulting copolymers are amphiphilic. The gradient monomer distribution allows us to tune the degree of hydrophobicity along the chain with a rather sharp transition between the hydrophobic and hydrophilic part in a hard gradient and a gradual change of hydrophobicity in a medium gradient.

**Ternary copolymers**. To expand the copolymerization and the resulting property library, we furthermore explored the copolymerization of three different cyclic phosphoesters. We selected a set of three monomers that should form relatively strong gradient structures with each other to result in amphiphilic structures resembling triblock terpolymers (Fig. 5). The influence of the organocatalyst on the resulting microstructure was further

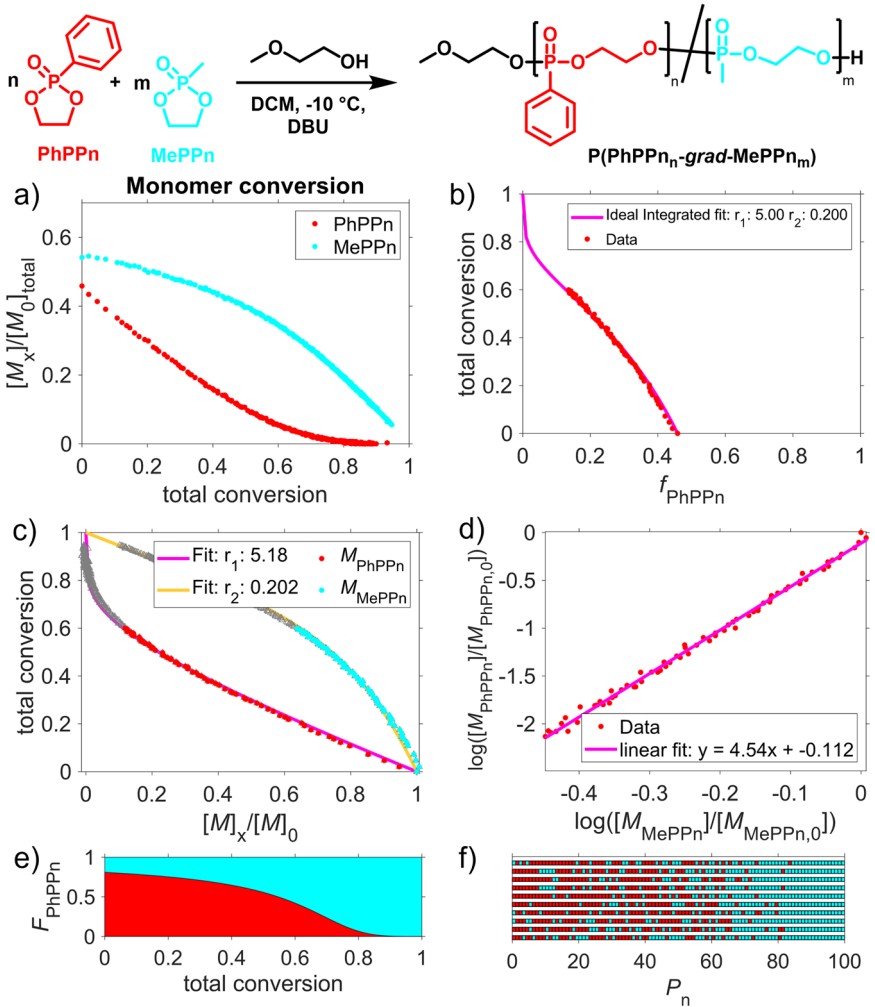

**Fig. 3 Copolymerization kinetics from real-time $^{31}$P NMR spectroscopy and calculation of reactivity ratios. a** Monomer concentrations as a function of the total conversion of 1 and 2 measured by $^{31}$P NMR spectroscopy (data obtained from the copolymerization of monomers **1** (PhPPn) and **2** (MePPn) (entry **1**, Table 1)); **b** Ideal integrated fit; **c** Jaacks fit; **d** BSL fit; for the fitting, data up to 60% of total conversion were used for all three models; **e** visualization of the copolymer compositions by a plot of the average monomer fraction composition against the total conversion; **f** Monte Carlo simulation using the determined reactivity ratios, showing the copolymers' microstructure (10 discrete polymer chains are shown).

investigated to additionally tune the kinetics. For self-assembly in water, asymmetric hydrophilic-hydrophobic-hydrophilic copolymer structure was targeted. Considering the gradient structures for the binary mixtures, the ternary mixture of EtPPn, ETP, and EPP was copolymerized using DBU/TU or DBU/tris Urea as the respective organocatalysts (**P9** and **P10**). These organocatalysts were chosen based on reported reactivities[14].

The copolymerization of three monomers with different catalysts allowed us to change the order of the blocks. Figure 5 summarizes the data calculated from the real-time $^{31}$P NMR spectroscopy kinetics, analyzed with ideal integrated, Jaacks, and BSL models (Figs. S11 and S12), provided six different reactivity ratios (Fig. 5d, h). For the copolymerization using DBU/TU, the order of monomer incorporation is EtPPn, ETP, and EPP, and it remains the same as in the binary system. Interestingly, the reactivity ratios of the individual monomer pairs during terpolymerization differed slightly from the ones calculated for the binary copolymers. Subsequently, we used the six averaged reactivity ratios to simulate the terpolymer structures. The simulation results were visualized as monomer fraction composition versus total conversion and Monte Carlo simulation of 10 discrete terpolymer chains (Fig. 5b, c). Although the reactivity ratios for the transition between the monomer blocks ($\Delta r_{12} = 14$

and $\Delta r_{23} = 13$) are not in the range of block-like structure, the Monte Carlo simulated polymer chains in Fig. 5c result in a triblock-like structure that appear to be a hard gradient type transitions.

To improve the reactivity as well as the total conversion, we used the more reactive TrisUrea co-catalyst (Fig. 5e)[14]. The total conversion of P10 was high, with 98% within 2 days (versus 5 days and 73% conv. with TU). The reactivity ratios changed compared to P9 (Fig. 5h). Interestingly, the incorporation preference for ETP and EPP are inverted for P10 compared to P9, which results in a different polymer composition. The monomer fraction composition versus total conversion as well as Monte Carlo simulated polymer chains reveal the hydrophilic-hydrophilic-hydrophobic structure P(EtPPn-*grad*-EPP-*grad*-ETP) (Fig. 5f, g). The monomer activation from the ThioUrea is less. Therefore, the ring-tension is still playing a big role in the reactivity; in comparison, the TrisUrea is strongly activating the monomer, and ring-tension plays less of a role. This strong activation is not given for the ETP due to the diffuse S = P double bond and the weak sulfur coordination by the urea protons[25]. The switch from TU to TrisUrea switched the incorporation order by changing the reactivity ratios from ETP and EPP while keeping EtPPn the fastest monomer.

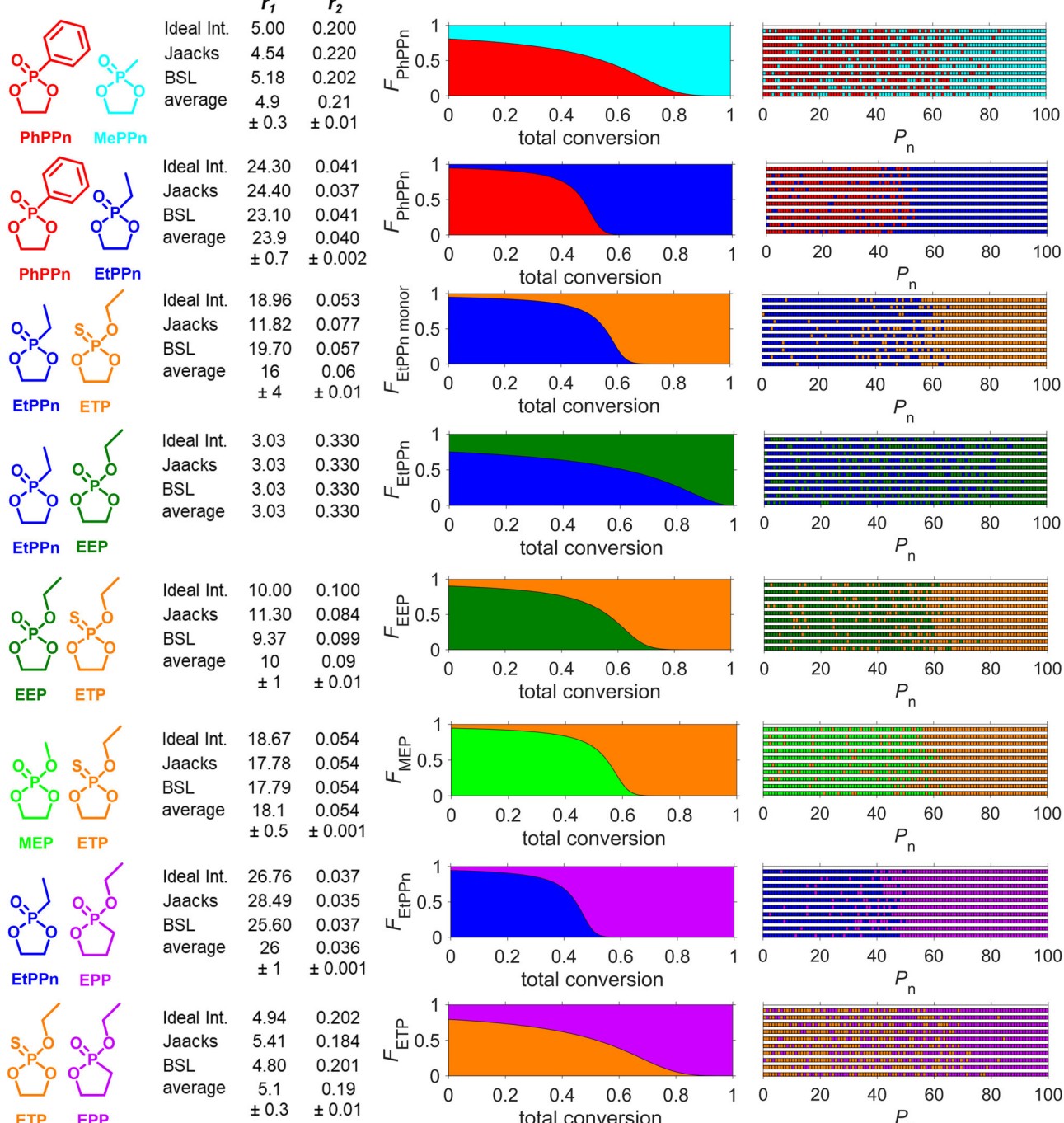

**Fig. 4 Overview of the copolymer microstructures of PPE copolymers synthesized in this study.** Each line shows the copolymerization of a set of two comonomers and the calculated reactivity ratios $r_1$ and $r_2$ using nonterminal models (ideal integrated, Jaacks, BSL, and the averaged values). The copolymer compositions are visualized by plotting the average monomer fraction composition against the total conversion (middle) and by 10 discrete copolymer chains calculated via Monte Carlo simulation using the determined reactivity ratios (right).

**Physical properties in water.** The potential applications of such binary and ternary copolymers should utilize their amphiphilic character, for example, to stabilize hydrophobic payload in biomedical context or to stabilize dispersions as surfactants[26–28]. To investigate the amphiphilic properties, interfacial tensions between water and cyclohexane of all polymers were measured using a spinning-drop tensiometer (Fig. 6a, Table S1). All copolymers were able to significantly reduce the interfacial tension to values below 22 mN/m. The determined surface tensions resulting from the two double hydrophilic copolymers (P4, P7) are higher

compared to the surface tensions determined from the other amphiphilic copolymers, as one would expect as they do not have hydrophobic blocks only some side-chain/main-chain amphiphilicity. All other copolymers with more hydrophobic segments displayed lower surface tensions (<14 mN/m), while the ETP-containing copolymers had the lowest values for the interfacial tension (11 mN/m); the ternary copolymers showed the lowest surface tensions (ca. 8 mN/m).

All the synthesized amphiphilic copolymers self-assembled in water, as shown by dynamic light scattering (DLS) (Fig. 6b).

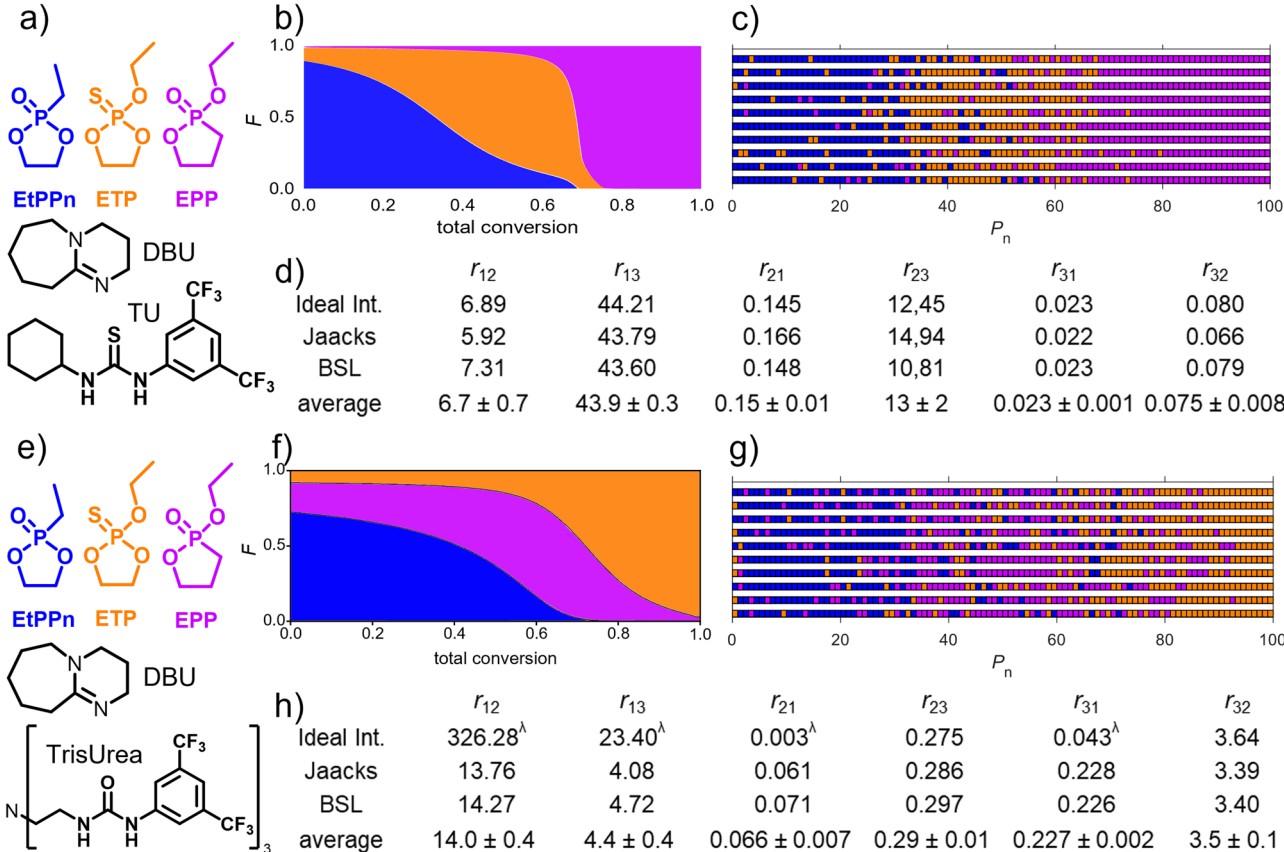

**Fig. 5 Overview of the microstructures of PPE terpolymers synthesized in this study with two different catalyst systems.** **a**, **e** The structure of the reaction components for the terpolymerization to P9 and P10. The reactivity ratios were calculated for pairs of two monomers, resulting in six reactivity ratios $r_{xy}$ and $r_{yx}$ using nonterminal models (ideal integrated, Jaacks, BSL, and the averaged values) for P9 (**d**) and P10 (**h**). The copolymer compositions are visualized by plotting the average monomer fraction composition against the total conversion (**b**) and (**f**) and by 10 discrete copolymer chains calculated via Monte Carlo simulation using the determined reactivity ratios (**c**) and (**g**) λ) these reactivity ratios were not used to calculate the average value.

Depending on the comonomer composition and the gradient strength, different nanostructures are obtained with diameters between ca 5 nm (presumably spherical micelles) up to ca 320 nm (presumably compound micelles). The DLS of ETP containing copolymers with EtPPn, EEP, and EPP (P3, P5, P8) yielded the hydrodynamic diameters ($D_h$) in a range of 150–320 nm suggesting self-assembly to colloidal structures, which is in line with lower interfacial tension (Table S1). The terpolymer P10 shows relatively small self-assembled colloidal structures of 38 nm due to the small volume fraction of the hydrophobic part. Overall, tailoring the microstructure and composition during the polymerization allowed us to change the surface tension and the size of the resulting colloids.

As PPEs have a high potential for the development of MRI-traceable polymer materials[17], we measured the NMR relaxation times ($T_1$ and $T_2$), which usually determine the signal intensity for MRI. Ideally, the $T_1$ relaxation time should be short, and $T_2$ should be long to gather images with high contrast. In Fig. 6c, the $T_1$ times of each comonomer in the corresponding copolymers are plotted. In general, the hydrophobic monomers show shorter relaxation times, similar to our earlier results[17]. As the hydrophobic part of the copolymer is expected to form a core of a micellar structure, the higher density of polymer chains in the assembled part leads to lower mobility and, consequently, to a faster relaxation[17]. Similarly, $T_2$ times are shown in Fig. 6d. The $T_1$, as well as $T_2$ relaxation time of EEP, varies a lot depending on the comonomer, as the P(EtPPn-*grad*-EEP) is fully dissolved, the better solubilization leads to slower

relaxation. While in P(EEP-*grad*-ETP), the ETP segment aggregates in water, and for the EEP, the local polymer density is increased as well. Therefore, shorter relaxation times are observed. The same effect can be detected for EPP, which gives in P(EtPPn-*grad*-EPP) the longest $T_2 = 1.74$ s times we measured. Our results suggest that $T_1$ and $T_2$ relaxation times can be tuned by the gradient strength (Table S1), which led to favorable MRI properties (Fig. 7). The MRI phantom images show these trends nicely, indicating a relatively high SNR of 171.4 for P(EtPPn-*grad*-EEP) followed by 23.9 for P(EtPPn-*grad*-EPP), 12.4 for P(EEP-*grad*-ETP), and 3.9 for P(ETP-*grad*-EPP) underlining the power of the PPE platform to develop MRI-traceable polymers for future diagnostics.

## Conclusion

Organocatalyzed ring-opening copolymerization of different cyclic phosphoesters gives access to a library of amphiphilic and potentially degradable copolymers that can act as MRI-traceable nanomaterials. For the first time, we report the real-time [31]P NMR kinetics of the copolymerization towards different polyphosphates, in- and side-chain polyphosphonates, and polythiophosphates. We prepared binary and ternary PPE copolymers and reported the respective reactivity ratios using different nonterminal models. With these data, the microstructures were calculated via Monte Carlo models, providing access to gradient copolymers with soft gradients ($\Delta r = 3$) up to block-like ($\Delta r = 26$) copolymers.

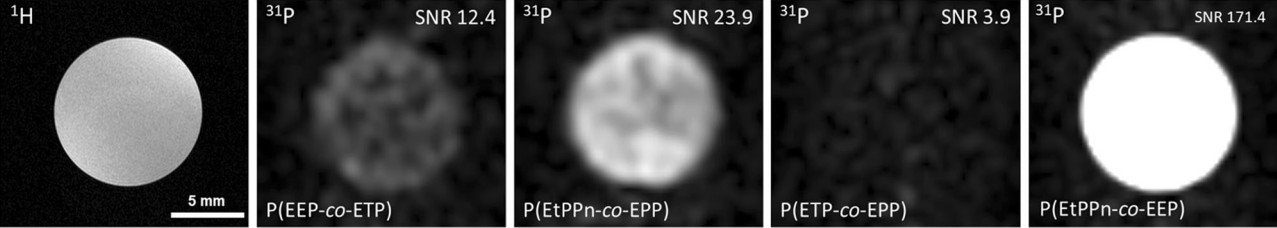

**Fig. 6 Physicochemical properties of the gradient copolymers P(X-grad-Y-grad-Z) prepared in this study. a** Interfacial tension water/cyclohexane was determined with the Laplace–Young's method for the concentrations 0.1, 1, and 10 g/L, the average value and standard deviation of the different concentrations are shown for each polymer, **b** hydrodynamic diameter and standard deviation (from three measurements) from self-assembled colloids for the polymers dissolved in water measured by DLS at 90° angle; relaxation times were measured in $H_2O:D_2O/9:1$ at 9.4 T (172 MHz for $^{31}P$) **c** shows T1 and **d** T2 relaxation times for each monomer in the gradient polymers.

**Fig. 7 MRI phantom measurements of the gradient polymers.** $^1H$ MRI image as reference and $^{31}P$ MRI images of three different gradient copolymers (from left to right P5, P7, P8, P4), the signals of both monomer-units of each polymer were added to a sum image, resulting in the highest signal to noise ratio (SNR). A phantom of 38 mg mL$^{-1}$ polymer solution was measured in a 9.4 T MRI spectrometer.

By adjusting the comonomer combination and the respective organocatalytic system, we synthesized amphiphilic copolymers while tailoring the gradient strength of hydrophilicity along the polymer chain. These copolymers are self-assembled into nanostructures and correlated to their NMR relaxation times depending on the local environment. Those polymers have a high potential for $^{31}P$ MRI-traceable nanomaterials, which are biocompatible and degradable based on the versatility of the PPE chemistry. Future work will investigate coatings for implants and drug encapsulation for theranostics, followed by $^{31}P$ MRI.

## Experimental
### Materials and methods
*Materials.* All solvents were purchased in HPLC grade or dry (purity > 99.8 %), and chemicals were purchased in the highest grade (purity > 98 %) from *Sigma Aldrich, Acros Organics, Fluka VWR chemicals*, or *Fisher Scientific* and used as received unless otherwise described. 1,8-Diazabicyclo[5.4.0]undec-7-ene (DBU) was distilled from calcium hydride and stored over molecular sieves (3 and 4 Å) under a nitrogen atmosphere. 2-(Benzyloxy) ethanol and 2-(methoxy)ethanol was purchased from *ABCR*, distilled from calcium hydride, and stored over molecular sieves

(4 Å) under a nitrogen atmosphere. Ethylene glycol was purchased dry and stored over molecular sieves (4 Å) under a nitrogen atmosphere. N-Cyclohexyl-N′-(3,5-bis(trifluoromethyl)phenyl)thiourea (TU) was synthesized according to the procedure described by Tripathi et al.[29] 1,1′,1″-(nitrilotris(ethane-2,1-diyl)) tris(3-(3,5-bis(trifluoromethyl)phenyl)urea) (TrisUrea) was synthesized according to the procedure described by Fastnacht et al.[25].

The syntheses of the monomers and the initial reports of their homo polymers can be found in the literature[6,9,14,15,18,30].

2-methyl-2-oxo-1,3,2-dioxaphospholane (MePPn) was synthesized according to the two-step procedure described by Steinbach et al.[9] 2-ethly-2-oxo-1,3,2-dioxaphospholane (EtPPn) was synthesized according to the two-step procedure described by Wolf et al.[6,18] 2-phenyl-2-oxo-1,3,2-dioxaphospholane (PhPPn) was synthesized according to the procedure described by Koshkina et al.[17] 2-ethoxy-1,2-oxaphospholane 2-oxide (EPP) was synthesized according to the procedure described by Bauer et al.[14] 2-methoxy-2-oxo-1,3,2-dioxaphospholane (MEP) and 2-ethoxy-2-oxo-1,3,2-dioxaphospholane (EEP) was synthesized according to the procedure described by Steinbach et al.[30] 2-ethoxy-2-thiono-l,3,2-dioxaphospholane (ETP) was synthesized according to the two-step procedure described by Liu et al[15]. The monomers were stored at −25 °C under a nitrogen atmosphere.

*Gel permeation chromatography.* GPC measurements were performed in DMF (containing $1 \text{ g L}^{-1}$ of LiBr) at 50 °C and a flow rate of $1 \text{ mL min}^{-1}$ with a *PSS SECcurity* as an integrated instrument, including three *PSS GRAM* column (30A/1000A/1000A) or a column set of two *PSS GRAM* (30A/1000A) and a refractive index (RI) detector. Calibration was carried out using poly(styrene) standards supplied by *Polymer Standards Service*. The SEC data were plotted with *OriginPro* 2019b software from *OriginLab* Corporation.

*Nuclear magnetic resonance (NMR) spectroscopy.* The [1]H and [31]P NMR spectra were measured on a 300, 400, 500, or 700 MHz *Bruker AVANCE III AMX* system or 600 MHz *Bruker AVANCE NEO* system. The temperature was kept at 298.3 K. As a deuterated solvent, DMSO-$d_6$, CDCl$_3$, or D$_2$O were used. For analysis of all measured spectra, *MestReNova* 9 from *Mestrelab Research S.L.* was used. The spectra were calibrated against the solvent signal.

*Magnetic resonance imaging.* MRI data were recorded at vertical Bruker AVANCE[III] and AVANCE NEO 9.4 T wide-bore NMR spectrometers driven by ParaVision 5.1 and 360 v3.2, respectively, and operating at frequencies of 400.2 MHz for [1]H and 162.0 MHz for [31]P measurements. Experiments were carried out using a Bruker microimaging unit (Micro 2.5) equipped with actively shielded gradient sets (capable of 1.5 T/m maximum gradient strength and 150 μs rise time at 100% gradient switching) and a dual tunable [1]H/[31]P 25-mm birdcage resonator. *[1]H and [31]P MRI of aqueous dispersion phantoms.* Aqueous dispersions were used at a concentration of polymer of $38 \text{ mg mL}^{-1}$. Multi-chemical selective imaging (mCSSI) was carried out as described previously[31] using selective exciting frequencies with a bandwidth of 913 Hz (gauss 3 ms): (TE 6.32 ms, TR 2.5 s, RARE factor 32, matrix 32 × 32, ST 8 mm, effective spectral bandwidth 15,000 Hz, NA 410, TAcq 17 min 5 s.

*Differential scanning calorimetry.* DSC measurements were performed using a Trios DSC 25 series thermal analysis system with a temperature range from −100 °C to 35 °C under nitrogen with a heating rate of $10 \text{ °C min}^{-1}$. All glass transition temperatures ($T_g$) were obtained from the second heating ramp of the experiment. The DSC data is reported in Fig. S2.

*Dynamic light scattering.* DLS was done at a Zetasizer Lab from Malvern, UK, at a scattering angle of 90° and 295 K. The samples were diluted with ultrapure water that the attenuator was at steps 10–11 (set automatically by the device). Data analysis was done with ZSxplorer 2.2.0.147 software from Malvern Analytical. Three measurements were done to determine an average value with a standard deviation.

*Spinning drop.* The interfacial tension was measured at 25 °C and 8000 rpm on a spinning drop video tensiometer from *dataphysics* temperature control via a liquid circulator (MC-TFC 25) running with the *SVT 20* software. Using Young–Laplace theory to calculate the interfacial tension.

## Polymer syntheses

*Representative procedure for the ring-opening copolymerization catalyzed with DBU (and cocatalysts).* Polymerization was performed according to modified literature protocols. The respective monomers and, if necessary, a cocatalyst (3.0 equivalents TU or TrisUrea with respect to the initiator) were weighed in a flame-dried Schlenk-tube, dissolved in dry benzene, and dried by lyophilization. The monomer was dissolved in dry dichloromethane to a total concentration of $4 \text{ mol L}^{-1}$. A stock solution of initiator 2-methoxyethanol (or 2-(Benzyloxy)ethanol) in dry dichloromethane was prepared with a concentration of $0.2 \text{ mol L}^{-1}$, and the calculated amount was added to the monomer solution. A stock solution of DBU in dry dichloromethane was prepared with a concentration of $0.2 \text{ mol L}^{-1}$. For the *real-time* NMR kinetics, an 0.7 mL aliquot of the monomer solution was transferred to an NMR tube. The monomer solution and the catalyst solution (DBU) were set to the respective reaction temperature (−10 °C). The polymerization was initiated by the addition of the calculated volume of catalyst solution containing 3.0 equivalents of DBU with respect to the initiator. Polymerization was terminated by the rapid addition of an excess formic acid dissolved in dichloromethane with a concentration of $20 \text{ mg mL}^{-1}$. The colorless, amorphous polymers were purified by precipitation into cold diethyl ether and dialyzed against pure ethanol overnight, and dried at reduced pressure. Yields ranged from 50 to 90%.

*Real-time NMR-kinetics of copolymerizations.* All measured parameters were determined on the reaction mixture as mentioned above (at −10 °C in CH$_2$Cl$_2$). The reaction was initiated by the addition of the calculated volume of a catalyst solution containing 3 eq. of DBU with respect to the initiator ($0.2 \text{ mol L}^{-1}$ in DCM). The tube was placed in the NMR spectrometer (at −10 °C), and [31]P NMR measurements were conducted. The delay time D1 was set to 30 s (the longest T1 was determined to be 9.2 s) and fast reaction spectra were generated with a single scan, while for slower reactions, the number of scans was increased to 4, 8, 16, or 32 scans.

Representative NMR data of P(PhPPn-*grad*-MePPn): [1]H NMR (CDCl$_3$, ppm): δ = 7.92–7.28 (m, phenyl protons), 4.49–3.92 (m, backbone -CH$_2$-), 3.30 (s, initiator -CH$_3$-O-), 1.56 (d, $^2J_{HP}$ = 18 Hz, P-CH$_3$).
[31]P NMR (CDCl$_3$, ppm): δ = 32.3 (MePPn), 19.8(PhPPn)

Representative NMR data of P(PhPPn-*grad*-EtPPn): [1]H NMR (CDCl$_3$, ppm): δ = 7.92–7.28 (m, phenyl protons), 4.49–3.87 (m, backbone -CH$_2$-), 3.30 (s, initiator -CH$_3$-O-), 1.90–1.52 (m, side-chain P-CH2-), 1.17 (d, $^3J_{HP}$ = 19 Hz, -CH$_3$).
[31]P NMR (CDCl$_3$, ppm): δ = 35.2 (EtPPn), 19.8(PhPPn)

Representative NMR data of P(EtPPn-*grad*-ETP): [1]H NMR (CDCl$_3$, ppm): δ = 4.48–4.01 (m, backbone -CH$_2$-, ETP

side-chain -O-CH$_2$-), 3.38 (s, initiator CH$_3$-O-), 1.81 (dq, $^2J_{HP}$ = 15.7 Hz, $^3J_{HH}$ = 7.7 Hz, side-chain P-CH$_2$-), 1.34 (t, $^3J_{HH}$ = 7.1 Hz, ETP side-chain -CH$_3$), 1.18 (dt, $^3J_{HP}$ = 20.5 Hz, $^3J_{HH}$ = 7.8 Hz, EtPPn side-chain -CH$_3$).
$^{31}$P NMR (CDCl$_3$, ppm): δ = 68.1 (ETP), 35.2 (EtPPn)

Representative NMR data of P(EtPPn-*grad*-EEP): $^1$H NMR (CDCl$_3$, ppm): δ = 4.35- 4.01 (m, backbone -CH$_2$-, EEP side-chain -O-CH$_2$-), 3.37 (s, initiator CH$_3$-O-), 1.81 (dq, $^2J_{HP}$ = 15.6 Hz, $^3J_{HH}$ = 7.5 Hz, side-chain P-CH$_2$-), 1.33 (t, $^3J_{HH}$ = 7.0 Hz, EEP side-chain -CH$_3$), 1.17 (dt, $^3J_{HP}$ = 20.6 Hz, $^3J_{HH}$ = 7.6 Hz, EtPPn side-chain -CH$_3$).
$^{31}$P NMR (CDCl$_3$, ppm): δ = 35.3 (EtPPn), −1.3 (EEP)

Representative NMR data of P(EEP-*grad*-ETP): $^1$H NMR (CDCl$_3$, ppm): δ = 4.44–4.00 (m, backbone -CH$_2$-, ETP and EEP side-chain --O-CH$_2$-), 3.38 (s, initiator CH$_3$-O-), 1.33 (m, ETP and EEP side-chain -CH$_3$),
$^{31}$P NMR (CDCl$_3$, ppm): δ = 68.1 (ETP), −1.3 (EEP)

Representative NMR data of P(MEP-*grad*-ETP): $^1$H NMR (CDCl$_3$, ppm): δ = 4.47–4.01 (m, backbone -CH$_2$-, ETP side-chains -O-CH$_2$- and MEP -O-CH$_3$), 3.38 (s, initiator CH$_3$-O-), 1.34 (q, $^3J_{HH}$ = 6.7 Hz ETP side-chain -CH$_3$),
$^{31}$P NMR (CDCl$_3$, ppm): δ = 68.1 (ETP), −1.3 (MEP)j

Representative NMR data of P(EtPPn-*grad*-EPP): $^1$H NMR (CDCl$_3$, ppm): δ = 7.34–7.30 (m, initiator benzylic protons), 4.55 (s, initiator benzyl-CH$_2$-O-), 4.32–3.96 (m, EtPPn backbone -CH$_2$-, EPP backbone P-CH$_2$-CH$_2$-C**H$_2$**-O/ side-chain -O-CH$_2$-), 2.01–1.90 (m, backbone P-CH$_2$-C**H$_2$**-), 1.80 (dq, $^2J_{HP}$ = 18.5 Hz, $^3J_{HH}$ = 7.7 Hz, side-chain P-CH$_2$-, EPP backbone P-C**H$_2$**-CH$_2$-), 1.31 (t, $^3J_{HH}$ = 7.0 Hz, EEP side-chain -CH$_3$), 1.16 (dt, $^3J_{HP}$ = 20.7 Hz, $^3J_{HH}$ = 7.6 Hz, EtPPn side-chain -CH$_3$).
$^{31}$P NMR (CDCl$_3$, ppm): δ = 35.3 (EtPPn), 34.8 (EtPPn), 31.9 (EPP), 31.5 (EPP)

Representative NMR data of P(ETP-*grad*-EPP): $^1$H NMR (CDCl$_3$, ppm): δ = 7.36–7.31 (m, initiator benzylic protons), 4.56 (s, initiator benzyl-CH$_2$-O-), 4.36–3.96 (m, ETP backbone -CH$_2$-, ETP side-chain -O-CH$_2$-, EPP backbone P-CH$_2$-CH$_2$-C**H$_2$**-O, EPP side-chain -O-CH$_2$-), 2.06–1.91 (m, backbone P-CH$_2$-C**H$_2$**-), 1.91–1.77 (m, EPP backbone P-C**H$_2$**-CH$_2$-), 1.39–1.26 (m, ETP,EEP side-chains -CH$_3$),
$^{31}$P NMR (CDCl$_3$, ppm): δ = 68.1 (EtPPn), 31.9 (EPP), 31.4 (EPP)

## Data and materials availability

All processed data are available in the manuscript or supplementary information. Raw data and materials are available upon request. Further NMR data is summarized in Supplementary Data 1.

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

## Acknowledgements
The authors thank Dr. Jens C. Markwart, Dr. Ricardo E.P. Martinho (UT), Bianca Ruel (UT), Clemens Padberg (UT), and Ramon ten Elshof (UT) for discussions and support in the analyses of the copolymers. O.K. thanks to the Alexander-von-Humboldt Stiftung for support.

## Author contributions
Investigation: T.R., F.R.W., and U.F. Supervision: O.K. and F.R.W. Writing—original draft: T.R. Writing—review & editing: T.R., O.K., F.R.W., and U.F.

## Competing interests
The authors declare no competing interests.
