## [Peer Review File · Communications Chemistry]

Reviewer #1 (Remarks to the Author):

The manuscript by Wurm and coworkers presented an interesting story about synthesis of polyphosphoesters in the presence of organic catalysts. The gradient copolymers with varying amphiphilicity and gradient strength were well produced by the combination of different comonomers and organocatalysts. These polymers have potential applications in drug and theranostics. Reactivity ratios were determined and simulated for different binary or/and ternary comonomers. Altogether, the scientific quality and the scholarly presentation of this manuscript are very good, thus the work is certainly of sufficient impact and novelty to warrant publication.

As a minor correction, I would suggest citing the following article on construction of polyphosphoesters with the main chain of rigid backbones and stereostructures via organocatalyzed ring-opening polymerization. *Polym. Chem.*, 2020, 11, 3475-3480.

Reviewer #2 (Remarks to the Author):

This paper is excellently written, and the results are clearly and systematically rationalized and explained. This class of copolymers grants access to a wide variety of properties, owing to the different reactivities afforded by the organocatalysis. I would recommend it for publication, essentially as is, with a couple exceptions.

Figure 2. Increase the font size of the axes.

Figure 6: It's a bit difficult to distinguish the colors. Consider (part c and d) making the symbols different and a bit larger size.

Reviewer #3 (Remarks to the Author):

The manuscript entitled "Real-time ^{31}P NMR of organocatalytic ring-opening copolymerization reveals different gradient strengths in polyphosphoesters – a road to MRI-traceable nanostructures" describes precise analytical data for copolymerization of various cyclic phosphorus-containing monomers and newly MRI properties of the copolymers. Overall, this manuscript is well written and contains important basic information to understand polymerization manner of cyclic phosphorus-containing monomers. The findings of this manuscript will be of use for the design new polymeric materials based on polyphosphoesters. The referee would like to just ask authors to add MRI image of P(EtPPn-co-EEP) to better understand the effect of structure of polymers on MRI properties.

Rebuttal

Communications chemistry

Reviewer #1 (Remarks to the Author):

The manuscript by Wurm and coworkers presented an interesting story about synthesis of polyphosphoesters in the presence of organic catalysts. The gradient copolymers with varying amphiphilicity and gradient strength were well produced by the combination of different comonomers and organocatalysts. These polymers have potential applications in drug and theranostics. Reactivity ratios were determined and simulated for different binary or/and trinary comonomers. Altogether, the scientific quality and the scholarly presentation of this manuscript are very good, thus the work is certainly of sufficient impact and novelty to warrant publication.

As a minor correction, I would suggest citing the following article on construction of polyphosphoesters with the main chain of rigid backbones and stereostructures via organocatalyzed ring-opening polymerization. *Polym. Chem.*, 2020, 11, 3475-3480.

We thank the reviewer for reading our manuscript and giving suggestions. We do not really see the connection of this publication (Zhen 2020) to our manuscript. Beside the fact that it discusses polyphosphoesters (PPE), no connection is apparent. We left out the suggested citation.

Reviewer #2 (Remarks to the Author):

This paper is excellently written, and the results are clearly and systematically rationalized and explained. This class of copolymers grants access to a wide variety of properties, owing to the different reactivities afforded by the organocatalysis. I would recommend it for publication, essentially as is, with a couple exceptions.

We thank the reviewer for reading our manuscript and giving constructive input to improve our manuscript.

Figure 2. Increase the font size of the axes.

Thanks a lot for the hint, we agree and changed the figure.

Figure 6: It's a bit difficult to distinguish the colors. Consider (part c and d) making the symbols different and a bit larger size.

Thanks for the remark, we increased the symbol size and changed the shape. The colors are adjusted as well.

Reviewer #3 (Remarks to the Author):

The manuscript entitled "Real-time ^{31}P NMR of organocatalytic ring-opening copolymerization reveals different gradient strengths in polyphosphoesters – a road to MRI-traceable nanostructures"

describes precise analytical data for copolymerization of various cyclic phosphorus-containing monomers and newly MRI properties of the copolymers. Overall, this manuscript is well written and contains important basic information to understand polymerization manner of cyclic phosphorus-containing monomers. The findings of this manuscript will be of use for the design new polymeric materials based on polyphosphoesters. The referee would like to just ask authors to add MRI image of P(EtPPn-co-EEP) to better understand the effect of structure of polymers on MRI properties.

We thank the reviewer for reading our manuscript and giving constructive input to improve our manuscript.

We agree on the suggestion, and added a MRI image of P(EtPPn-grad-EEP) in Figure 7.